# OpenReview forum: "The Reasoning Trap: How Enhancing LLM Reasoning Amplifies Tool Hallucination"
_ICLR.cc/2026/Conference — ICLR 2026 Conference Withdrawn Submission_

### Official Review · Reviewer_Zskr · 2025-10-28

**Soundness:** 2
**Presentation:** 1
**Contribution:** 2
**Rating:** 2
**Confidence:** 2

**Summary:**

This paper investigates whether enhancing the reasoning capabilities of LLMs increases tool hallucination. By introducing the SIMPLETOOLHALLUBENCH benchmark, the study finds that RL significantly increases tool hallucination rates while improving task performance. This phenomenon is not limited to specific training data or methods but also appears across different models and training approaches.

**Strengths:**

Novelty: The first systematic exploration of the causal relationship between reasoning enhancement and tool hallucination, filling a gap in the field.

Experimental Design: The use of the SIMPLETOOLHALLUBENCH benchmark clearly isolates the impact of different failure modes.

**Weaknesses:**

The conclusion that reasoning enhancement leads to hallucination seems unsurprising and uninteresting.

The paper only tests a small subset of models, with the largest parameter count being 32B, which makes its conclusions appear less reliable.

The overall structure is loose and inappropriate; such as Table 1, Table 2, and formulas (1) and (2). The layout of different section makes it a bit difficult to read.

The use of "DeepSeek-R1 as the judge" is uncommon, and no sufficient justification is provided.

**Questions:**

I regret that I failed to see the unique advantages of this benchmark. For instance, could the use of this benchmark lead to conclusions that contradict those derived from other benchmarks(such as API-Bank)? Or does it merely test a different yet similar capability, ultimately yielding the same conclusion? I request the authors to provide a thorough explanation, supported by relevant citations. If the authors can fully address these concerns in their response, I would be willing to increase the score.

---

> ### Author Response · Authors · 2025-11-20
> **Author Response to Reviewer Zskr(Part 1)**
>
> We thank the reviewer for their valuable feedback. Our responses to the specific concerns are detailed below:
>
> **For `Weakness 1`**: We understand the reviewer’s concern. However, **recent works show that the relationship between reasoning and hallucination is neither trivial nor settled**, especially for reasoning and agentic LLMs. Even for standard factual hallucination, the relationship between reasoning and reliability is not considered settled. Recent studies explicitly show that stronger reasoning can increase hallucination (e.g., the o3/o4-mini system card reporting “more claims overall” leading to more incorrect assertions) while others show that reasoning can hurt reliability recall at key operating points (Reasoning’s Razor [2], Yao et al. [3]). Meanwhile, Li et al. [4] demonstrate that more capable LLMs are not uniformly safer, Safety Tax [5] shows that aligning safety can reduce reasoning ability, and SafePath [6] documents trade-offs when suppressing harmful chain-of-thought. These contradictory findings show that even for conventional hallucination, **the effect of reasoning remains an active research question**, not an obvious fact. All above papers are accepted by top tier conferences.
>
> As RL-based agents[8,9,10,11] become increasingly capable, tool hallucination becomes even more critical: unlike textual hallucinations, **a fabricated or misused tool call directly triggers incorrect API actions and can break real systems**. Yet this failure mode has not been systematically studied. Our work provides the first **causal, mechanistic**, and **tool-specific** analysis showing how reasoning enhancement amplifies tool hallucination. Given both the scientific uncertainty in prior literature and the practical stakes for agent systems, establishing this relationship clearly is both novel and important.
>
> **For `Weakness 2`**: We agree that evaluating only up to 32B parameters in our RL experiments is a limitation; this stems from the substantial computational cost of running full GRPO-style reasoning RL. We can try our best to mitigate this concern by expanding the empirical section with an additional table that evaluates much larger proprietary reasoning models and their corresponding non-reasoning counterparts on SimpleToolHalluBench. Concretely, we compare (i) Qwen3-235B-A22B-Instruct-2507 vs. Qwen3-235B-A22B-Thinking-2507, (ii) DeepSeek-V3-671B vs. DeepSeek-R1-671B, and (iii) Kimi-K2-1T vs. Kimi-K2-Thinking-1T, reporting hallucination rates under both NTA and DT settings:
>
> | Model                         | $R_{NTA}(\\%)$ | $R_{DT}(\\%)$ |
> | ----------------------------- | ------------- | ------------ |
> | Qwen3-235B-A22B-Instruct-2507 | 3.7           | 23.3         |
> | Qwen3-235B-A22B-Thinking-2507 | 6.1           | 30.7         |
> |                               |               |              |
> | DeepSeek-V3-671B              | 10.8          | 33.8         |
> | DeepSeek-R1-671B              | 17.6          | 42.6         |
> |                               |               |              |
> | Kimi-K2-1T                    | 1.0           | 15.5         |
> | Kimi-K2-Thinking-1T           | 4.4           | 21.3         |
>
> Across all three pairs we observe the same qualitative pattern as in the smaller open-source models: the “thinking”/reasoning model consistently **increases tool hallucination rates** on SimpleToolHalluBench. This suggests that the reasoning–hallucination trade-off we document is not an artifact of 7B–32B backbones, but **persists even for models in the 200B–1T range**.
>
>
> **For `Weakness 3`**: We appreciate the reviewer for pointing out issues with the current layout. We will revise the manuscript to streamline the exposition: merging formulas (1) and (2), and reorganizing the sections and images for better reading.

---

> ### Author Response · Authors · 2025-11-20
> **Author Response to Reviewer Zskr(Part 2)**
>
> **For `Weakness 4`**: LLM-as-a-judge has rapidly become a standard evaluation paradigm, with **recent work explicitly proposing reasoning models as judges** and surveying their growing adoption [12,13]. Within this paradigm, **DeepSeek-R1 has emerged as a strong open-source large reasoning model and a de facto judge baseline**: both J1 [12] and JudgeLRM [13] explicitly adopt DeepSeek-R1 as a core LLM-as-a-judge baseline, reflecting its status as a competitive and practically useful judge model. Empirically, DeepSeek-R1 attains near state-of-the-art results on MMLU, MMLU-Pro, and GPQA Diamond, and **often matches or even surpasses closed-source models such as GPT-4o/OpenAI-o1** [15]. Moreover, practitioner reports and open-source toolkits increasingly treat DeepSeek-R1 as a practical judge/teacher model (e.g., for reward modeling and evaluation) because it combines strong reasoning ability with relatively low inference cost [12,13,15].
> In our setting, these properties make DeepSeek-R1 a natural choice: it is (i) competitive with proprietary models, (ii) fully open and reproducible, and (iii) inexpensive enough to deploy at scale. While we acknowledge that DeepSeek-R1 is currently less historically established as a judge than GPT-4–family models, we deliberately choose it because it **offers state-of-the-art reasoning and judgment performance at much lower cost**, and existing evaluations show its judge performance to be fully competitive with GPT-4–family models while remaining reproducible and affordable at scale [15].
>
> **For `Question`**: Existing tool-augmented and agent benchmarks such as API-Bank [17], ToolBench and its successors [18], and AgentBench [19] are primarily designed to measure **capability**: they focus on whether an agent can plan, select tools, and complete tasks when the necessary tools are available and appropriate. In our work, however, the central conclusions do not follow from such capability benchmarks. Analogously, in the traditional LLM setting, there has long been a large ecosystem of benchmarks for general language understanding and QA, **yet this has never prevented dedicated hallucination benchmarks** (e.g., HalluLens [14], MIRAGE-Bench [16], HaluEval[20]) from being proposed and accepted at top venues, because they **capture a distinct dimension of reliability rather than duplicating capability evaluation**.
>
> SimpleToolHalluBench plays the same role for tool-using agents. As LLM agents with agentic RL[8,9,10,11] become the dominant deployment form, their tool-calling capabilities are already carefully evaluated by API-style and agent benchmarks, but **there is still no benchmark that can support the key conclusions of this paper**: that strengthening reasoning (e.g., via RL) can systematically amplify tool hallucination and induce a sharp capability–reliability trade-off. Existing capability benchmarks will only show that “the agent gets better at using tools,” whereas SimpleToolHalluBench is purpose-built to probe tool hallucination under controlled tool availability—no-tool and distractor-tool settings where the task is impossible to solve correctly and the only reliable behavior is honest abstention. In this regime, we explicitly observe cases where reasoning-oriented RL improves task rewards on tool-reasoning benchmarks but simultaneously worsens performance on our reliability-focused instances, a pattern that capability-centric benchmarks like API-Bank [17] are not designed to reveal. We therefore view SimpleToolHalluBench as complementary rather than redundant: it fills a missing piece by providing a targeted, scalable, and controllable way to study tool-specific hallucination and the resulting system-level failures in modern think-then-act agents[8,9,10,11].

---

> ### Author Response · Authors · 2025-11-20
> **Author Response to Reviewer Zskr(Part 3)**
>
> **References**
>
> [1] OpenAI. (2025). *OpenAI o3 and o4-mini system card*. OpenAI.
>
> [2] Chegini, A., Kazemi, H., Souza, G., Safi, M., Song, Y., Bengio, S., Williamson, S., & Farajtabar, M. (2025). *Reasoning’s Razor: Reasoning improves accuracy but can hurt recall at critical operating points in safety and hallucination detection*. arXiv preprint arXiv:2510.21049.
>
> [3] Yao, Z., Liu, Y., Chen, Y., Chen, J., Fang, J., Hou, L., Li, J., & Chua, T.-S. (2025). *Are reasoning models more prone to hallucination?* arXiv preprint arXiv:2505.23646.
>
> [4] Li, A., Mo, Y., Li, M., Wang, Y., & Wang, Y. (2025). *Are smarter LLMs safer? Exploring safety–reasoning trade-offs in prompting and fine-tuning*. arXiv preprint arXiv:2502.09673.
>
> [5] Huang, T., Hu, S., Ilhan, F., Tekin, S. F., Yahn, Z., Xu, Y., & Liu, L. (2025). *Safety tax: Safety alignment makes your large reasoning models less reasonable*. arXiv preprint arXiv:2503.00555.
>
> [6] Jeung, W., Yoon, S., Kahng, M., & No, A. (2025). *SafePath: Preventing harmful reasoning in chain-of-thought via early alignment*. Neurips 2025.
>
> [7] Akbar, S. A., Hossain, M. M., Wood, T., Chin, S.-C., Salinas, E. M., Alvarez, V., & Cornejo, E. (2024). HalluMeasure: Fine-grained hallucination measurement using chain-of-thought reasoning. EMNLP 2024.
>
> [8] Chen, M., Li, T., Sun, H., Zhou, Y., Zhu, C., Yang, F., Zhou, Z., Chen, W., Wang, H., Pan, J. Z., Zhang, W., & Chen, H. (2025). *ReSearch: Learning to Reason with Search for LLMs via Reinforcement Learning*. Neurips 2025.
>
> [9] Jin, B., Zeng, H., Yue, Z., Yoon, J., Arik, S. O., Wang, D., Zamani, H., & Han, J. (2025). *Search-R1: Training LLMs to Reason and Leverage Search Engines with Reinforcement Learning*. COLM 2025.
>
> [10] Xi, Z., Huang, J., Liao, C., Huang, B., Guo, H., Liu, J., Zheng, R., Ye, J., Zhang, J., Chen, W., He, W., Ding, Y., Li, G., Chen, J., Gui, T., Wu, Z., Zhang, Q., Huang, X., & Jiang, Y.-G. (2025). *AgentGym-RL: Training LLM Agents for Long-Horizon Decision Making through Multi-Turn Reinforcement Learning*. arXiv preprint arXiv:2509.08755.
>
> [11] Feng, J., Huang, S., Qu, X., Zhang, G., Qin, Y., Zhong, B., Jiang, C., Chi, J., & Zhong, W. (2025). *ReTool: Reinforcement Learning for Strategic Tool Use in LLMs*. arXiv preprint arXiv:2504.11536.
>
> [12] Whitehouse, C., Wang, T., Yu, P., Li, X., Weston, J., Kulikov, I., & Saha, S. (2025). *J1: Incentivizing thinking in LLM-as-a-judge via reinforcement learning*. arXiv preprint arXiv:2505.10320.
>
> [13] Chen, N., Hu, Z., Zou, Q., Wu, J., Wang, Q., Hooi, B., & He, B. (2025). *JudgeLRM: Large reasoning models as a judge*. arXiv preprint arXiv:2504.00050.
>
> [14] Bang, Y., Ji, Z., Schelten, A., Hartshorn, A., Fowler, T., Zhang, C., ... & Fung, P. (2025). Hallulens: Llm hallucination benchmark. *arXiv preprint arXiv:2504.17550*.
>
> [15] Guo, D., DeepSeek-AI, et al. (2025). *DeepSeek-R1: Incentivizing reasoning capability in LLMs via reinforcement learning*. arXiv preprint arXiv:2501.12948.
>
> [16] Thakur, N., Kazi, S., Luo, G., Lin, J., & Ahmad, A. (2025, April). Mirage-bench: Automatic multilingual benchmark arena for retrieval-augmented generation systems. NAACL 2025.
>
> [17] Li, M., Zhao, Y., Yu, B., Song, F., Li, H., Yu, H., Li, Z., Huang, F., & Li, Y. (2023). API-Bank: A comprehensive benchmark for tool-augmented LLMs. EMNLP 2023.
>
> [18] Xu, Q., Zhang, Z., Dong, Y., Wang, C., Chen, Y., & Xu, B. (2023). On the tool manipulation capability of open-source large language models. arXiv preprint arXiv:2305.16504.
>
> [19] Liu, X., Yu, Y., et al. (2023). AgentBench: Evaluating LLMs as agents. arXiv preprint arXiv:2308.03688.
>
> [20] Li, J., Cheng, X., Zhao, W. X., Nie, J. Y., & Wen, J. R. (2023). HaluEval: A Large-Scale Hallucination Evaluation Benchmark for Large Language Models. EMNLP2023.

---

> ### Author Response · Authors · 2025-11-26
> **Kind Request for Your Response**
>
> Dear Reviewer Zskr,
>
> As the discussion period is approaching its end, I would like to confirm that we have adequately addressed all of your concerns. If there are any additional comments or suggestions you would like us to consider, please do not hesitate to let us know. Your feedback is extremely valuable to us, and we remain eager to resolve any remaining issues and further improve the paper.
>
> Thank you again for your time and effort in reviewing our work.

---

### Official Review · Reviewer_4Xi5 · 2025-10-30

**Soundness:** 3
**Presentation:** 3
**Contribution:** 3
**Rating:** 4
**Confidence:** 3

**Summary:**

This paper investigates why reasoning reinforcement learning (Reasoning RL) often causes tool hallucination in large language models. Using Qwen2.5-7B-Instruct as the base model, the authors compare two RL frameworks: ReCall, which is tool-based RL trained on SynTool, and GRPO, a reasoning-focused RL trained on GSM8K. They find that Reasoning RL substantially increases the rate of tool hallucination, even when the model has never been exposed to tool-use data. Mechanistic analysis reveals two main effects: representation collapse, where post-RL models show reduced layerwise similarity to the pre-RL baseline on tool tasks, indicating disruption of tool-related representations; and residual stream amplification, where differences between correct and hallucinated behaviors accumulate gradually in later residual streams rather than emerging from a discrete failure. To mitigate hallucination, the authors test prompt engineering and preference optimization (DPO). Explicitly instructing the model not to use unavailable tools has minimal impact, while DPO significantly reduces hallucination but also weakens its tool-use ability. Overall, the study highlights an inherent reliability–capability trade-off in aligning reasoning-optimized models for tool use.

**Strengths:**

The paper tackles a timely and important question: how reasoning-oriented reinforcement learning (Reasoning RL) amplifies tool hallucination in large language models. It combines behavioral experiments, controlled RL training, and mechanistic analyses (CKA and residual-stream separability) to build a coherent causal story. The work is methodologically clear, empirically solid, and uses open models and benchmarks, enhancing reproducibility. Its identification of late-layer residual streams as loci of hallucination divergence provides novel mechanistic insight, and the study’s framing of a reasoning–reliability trade-off offers conceptual value for future alignment and safety research.

**Weaknesses:**

**1. Lack of cross-framework and cross-domain validation (Section 4)**:

The experimental design in §4.1–4.2 couples dataset and RL framework, where ReCall is only applied to SynTool and GRPO only to GSM8K. This coupling makes it difficult to determine whether the increase in tool hallucination stems from the reasoning RL framework or the used datasets. If possible, a full 2×2 design (SynTool × GRPO and GSM8K × ReCall) would provide a more rigorous disentanglement of (a) task domain effects and (b) RL framework biases.

**2. Potential domain-asymmetry in representation analysis (Section 5.1)**:

The CKA results are computed only on the GRPO-trained model, where the “in-distribution” is a reasoning task (GSM8K) and the “out-of-distribution” is tool usage. What about GRPO on the tool usage task and calculate the CKA results on both the reasoning task (ood) and the tool task (in-distribution)?

**3. Ambiguity in interpreting the residual-stream signal (Section 5.2)**:

The reported discrimination score (>0.14) for the residual stream is not large in absolute magnitude, and the residual stream itself is the aggregation of both the attention and MLP outputs.

Also, the analysis is correlational: it identifies where divergence manifests, but not what causes it. Further mechanistic verification (e.g., activation patching) would substantiate the claim that hallucination behavior originates from late-layer residual dynamics.

**Questions:**

**1. Cross-framework consistency:**

Have you tested whether applying GRPO on SynTool or ReCall on GSM8K yields similar hallucination amplification trends?
This would confirm whether the effect is framework-agnostic or contingent on specific algorithmic biases (e.g., reward shaping differences between GRPO and ReCall).

**2. Symmetry in representation collapse:**

In §5.1, representation collapse is shown only for non-agentic RL (GSM8K + GRPO).
If a tool-specific RL model (ReCall-trained on SynTool) were analyzed with the same CKA pipeline, would the collapse pattern invert (stable for tool-domain, unstable for reasoning-domain)?
Such a symmetric test could verify whether tool representations are intrinsically fragile or if the phenomenon simply reflects domain shift.

**3. Interpretation of the residual-stream discrimination score:**

Since the residual stream equals the running sum of attention and MLP outputs, how should we interpret its higher separability (>0.14)?
Does it indicate that hallucination originates in the residual flow, or merely that this channel amplifies upstream micro-differences?Have you considered “delta-residual” or “activation patching” analyses to verify causal attribution?

**4. On scale and significance of discrimination score:**
Given that the discrimination difference (≈0.07 vs 0.14) is moderate, is it statistically significant across random seeds or sample batches?
Providing standard deviations or confidence intervals would help readers gauge robustness.

**5. Interaction with model size and types:**
Did you observe similar trends (representation collapse or residual amplification) across different backbone sizes (e.g., Qwen2.5-7B vs 14B) and across normal and reasoning models (e.g., Qwen2.5-7B vs deepseek r1)?

---

> ### Author Response · Authors · 2025-11-20
> **Author Response to Reviewer 4Xi5(Part 1)**
>
> We sincerely appreciate the reviewer’s time and constructive comments. We have carefully considered the concerns raised and provide the following clarifications:
>
> **For `Weakness 1` and `Question 1`**:  Thanks for the suggestions. We would like to clarify that **ReCall is not a new RL algorithm, but a training framework whose underlying optimization method is GRPO**. Consequently, both section 4.1 (SynTool + ReCall) and section 4.2 (GSM8K + GRPO) rely on the same GRPO as the RL algorithm, and differ only in their training domains (tool use vs. math reasoning). As our goal is to study the relationship between hallucination and utility, it is important to keep the optimization framework consistent across experiments. Regarding cross-domain validation, since ReCall is essentially GRPO, the results from SynTool + ReCall and GSM8K + GRPO already constitute cross-domain validation. We apologize for the earlier misunderstanding and will explicitly clarify in the revised paper that ReCall is a GRPO-based training pipeline.
>
> **For `Weakness 2` and `Question 2`**: Following your suggestions, we additionally conduct the experiments to  compute CKA between the base model and the SynTool+ReCall(GRPO)-trained model, evaluating on both tool inputs and GSM8K reasoning inputs (**See Figure 7 in the Appendix D.2**). The resulting curves show a similar pattern to Fig. 4: for both tool and reasoning queries, CKA drops sharply in the first few layers (down to ≈0.86–0.88), with only a modest gap between the two domains. In particular, we do **not** see an inverted pattern where tool representations remain almost unchanged while reasoning collapses; instead, both domains exhibit comparable early-layer drift, and their CKA trajectories largely track each other. Crucially, the GRPO+SynTool agent RL setting does *not* yield a “safe” in-distribution regime for tool queries: despite being the direct training domain, tool inputs still undergo substantial representational drift relative to the base model. This indicates that RL focused solely on tool utility does not alleviate reliability issues; it can actually push tool-reliability representations away from the base along the very dimensions where tool-reliability queries (such as SimpleToolHalluBench) become more fragile.

---

> ### Author Response · Authors · 2025-11-20
> **Author Response to Reviewer 4Xi5(Part 2)**
>
> **For `Weakness 3` and `Question 3&4`**:  We thank the reviewer for noticing a mistake in the originally reported numbers: the average MLP discrimination scores should be **below 0.04**, and we have corrected this in the revised version. Our “discrimination score” is actually computed as the improvement over random guess (0.5) in a balanced binary classification of correct vs. hallucinated trajectories using a linear probe. A residual-stream score of ≈0.14 therefore corresponds to about **64% accuracy**, whereas attention and MLP outputs remain near below 0.02 and 0.04 (≈52–54% accuracy). Thus, late-layer residual representations offer roughly **triple the gain over chance** compared to individual attention/MLP components, and this advantage appears consistently in layers 20 and beyond. We agree that this does not imply the residual stream is the sole causal origin of hallucination; rather, it indicates that the residual stream is where the cumulative effect of many small upstream differences becomes most linearly readable. This is compatible with viewing the residual as the model’s central “workspace” that aggregates attention and MLP contributions over depth. This view is consistent with existing mechanistic interpretability work that models the residual stream as a shared communication channel to which layers read and write information [1], and with activation-steering methods for hallucination reduction that intervene by adding steering vectors to mid- or late-layer residual/MLP activations to improve truthfulness and reduce hallucinations [2,3].
>
> **To assess robustness, we trained the linear probes with five different random seeds**; in layers 20 and above, the residual-stream discrimination score is $0.152 \pm 0.011$, indicating that the effect is statistically stable across probe initializations and sample batches. More causal analyses are a promising direction for future work and complementary to our current correlational probes.
>
> **For `Question 5`:** Our CKA analysis is inherently cross-model between the reasoning and normal models: it always compares a **normal** base model to its **reasoning-RL-trained** counterpart, thereby directly quantifying representational drift between standard and reasoning models on both non-tool reasoning(math) and tool-reliability benchmarks. Due to computational limits, we did not run full GRPO training then CKA analysis for larger backbones such as 14B or 32B models. However, section 4.3 already provides size- and architecture-diverse *behavioral* evidence. For Qwen3-8B and Qwen3-32B, simply enabling the built-in “thinking” mode leads to higher tool hallucination rates on SimpleToolHalluBench (both NTA and DT), which indicates that a think–reliability trade-off emerges consistently across different backbone sizes.
>
> ---
>
> **References**
>
> [1] Dar, G., Xu, S., Wang, Y., Ravfogel, S., & Goldberg, Y. (2023). *Analyzing transformers in embedding space*. ACL 2023.
>
> [2] Rimsky, N., Belrose, N., & Radhakrishnan, A. (2024). *Steering Llama 2 via contrastive activation addition*. ACL 2024.
>
> [3] Wang, T., Liu, S., Fu, D., Song, Y., Zhang, C., & Wang, B. (2025). *Adaptive activation steering: A tuning-free LLM truthfulness improvement method for diverse hallucination categories*. WWW 2025.

---

> ### Author Response · Authors · 2025-11-26
> **Kind Request for Your Response**
>
> Dear reviewer 4Xi5,
>
> We sincerely appreciate the time and effort you have dedicated to reviewing our paper. Our reply has been available for several days now, yet we have not received any response from you. We respectfully inquire if any additional information requires clarification regarding our submission. Your prompt response would be greatly appreciated. Thank you for your valuable contribution to our research.
>
> Thank you for your time and effort in reviewing our paper.

---

### Official Review · Reviewer_rLkj · 2025-11-02

**Soundness:** 3
**Presentation:** 4
**Contribution:** 3
**Rating:** 6
**Confidence:** 4

**Summary:**

The paper systematically studies if improving LLMs’ reasoning ability induces more tool hallucination and gives affirmative answer by several experimental studies. It first introduces a new benchmark the SimpleToolHalluBench to study if the LLM will hallucinate under settings with no tools or distractor tools. Using this benchmark, the authors experimentally shows that the task performance increases but so does tool hallucination. Next, the paper delivers a mechanistic analysis via standard metrics like Centered Kernel Alignment, showing that RL reasoning training will preserve representations relevant to the training task while destabilizing those for tool use. Finally, the paper confirms that there's no free lunch in such reliability-capability trade-off since both prompt engineering and DPO reduces hallucination only by hurting tool-use ability.

**Strengths:**

- The paper is clearly written and easy to understand.
- The experimetal results suppoort the claim from different perspectives.

**Weaknesses:**

- The experimental results are solid, but it will be better to provide theoretical insights for the studied reliability-capability trade-off.
- In Section 4.3, more reasoning models can be tested on SimpleToolHalluBench to strengthen the claim.

**Questions:**

- Why does the paper limit the scope on tool hallucination only? I think the similar experiment can be done under other settings to study such reliability-capability trade off more systematically.

---

> ### Author Response · Authors · 2025-11-20
> **Author Response to Reviewer rLkj(Part 1)**
>
> Thank you for your time and effort in reviewing our paper, as well as your recognition of the strengths in our work. In response to the weaknesses and suggestions you raised, we respectfully provide the following clarifications:
>
> **For `Weakness 1`**:  The reliability–capability trade-off we observe can be understood through a shared-representation perspective. Concretely, the layers of the model implement a shared residual representation $h(x;\theta)$ on top of which multiple “heads” act: reasoning heads $w_{\text{reasoning}}$ that are directly optimized by Reasoning RL, and tool heads $w_{\text{tool}}$ that govern tool-usage decisions. Reasoning RL shapes $\theta$ so that $h(x;\theta)$ aligns with $w_{\text{reasoning}}$ and improves the margin for the reasoning ability boundary. When $w_{\text{reasoning}}$ and $w_{\text{tool}}$ are not perfectly orthogonal, pushing representations toward the reasoning-optimal subspace inevitably perturbs the activations relevant for tool reliability. **Intuitively, this is a standard multi-task interference effect: the same shared features cannot simultaneously maximize both reasoning and tool-reliability margins when their decision boundaries differ.**
>
> Behaviorally, this makes the model increasingly inclined, during its reasoning process, to “find a way” to produce an answer or invoke tools that superficially satisfy the user’s query, even in cases where it ought to recognize that the query cannot be reliably solved or should result in a conservative failure/abstention. This mechanism is consistent with our CKA and linear-probe analyses, which show both (i) a collapse of residual-stream similarity between the base and reasoning-tuned models and (ii) degraded linear separability of tool pathways. It also aligns with recent work showing that stronger reasoning can systematically trade off against safety-oriented metrics under shared parameters and objectives [1,2,3]. Together, these results provide a conceptual explanation: when reasoning and reliability objectives cohabit the same representation space, optimizing one can mechanically distort the other.
>
> **For `Weakness 2`**:  Following the reviewer’s suggestion, we will extend Section 4.3 by evaluating additional reasoning models on SimpleToolHalluBench. We now report results for multiple open-source reasoning-tuned LLMs trained with different RL or instruction-tuning recipes, and summarize their performance in a new table in the revised manuscript. For each model, we measure the tool hallucination rates under the same NTA/DT protocols as in the main text. The new experiments show a consistent trend: models with stronger reasoning generally exhibit higher tool hallucination rates on our benchmark, reinforcing our main claim that improving reasoning can worsen tool reliability.
>
> | Model                           | $R_{NTA}(\\%)$ | $R_{DT}(\\%)$ |
> | ------------------------------- | ------------- | ------------ |
> | Llama3.1-8B-Instruct            | 62.5          | 99.7         |
> | DeepSeek-R1-Distill-Llama3.1-8B | 96.3          | 100          |
> |                                 |               |              |
> | Qwen3-4B-Instruct-2507          | 3.4           | 24.0         |
> | Qwen3-4B-Thinking-2507          | 29.4          | 32.1         |
> |                                 |               |              |
> | Qwen3-235B-A22B-Instruct-2507 | 3.7           | 23.3         |
> | Qwen3-235B-A22B-Thinking-2507 | 6.1           | 30.7         |
> |                               |               |              |
> | DeepSeek-V3-671B              | 10.8          | 33.8         |
> | DeepSeek-R1-671B              | 17.6          | 42.6         |
> |                               |               |              |
> | Kimi-K2-1T                    | 1.0           | 15.5         |
> | Kimi-K2-Thinking-1T           | 4.4           | 21.3         |

---

> ### Author Response · Authors · 2025-11-20
> **Author Response to Reviewer rLkj(Part 2)**
>
> **For `Question 1`**:  In this work, reliability is considered mainly along two practically important axes: safety and hallucinations. Recent studies already document reasoning–safety trade-offs, showing that enhanced reasoning can degrade safety metrics or incur a “safety tax” on reasoning ability in safety-aligned models [1,2,3]. By contrast, hallucination-oriented work has largely focused on **factual** hallucinations in summarization and QA, including fine-grained hallucination measurement using chain-of-thought [4], but has not examined tool hallucinations in agentic settings. Our motivation is that tool-augmented agents are becoming central in practice, and “think-then-react” paradigms—where the model first reasons and then decides which tools to call—are now standard, yet the impact of stronger reasoning on **tool-centric reliability** has not been systematically studied. Our contribution is to fill this gap: we provide, to our knowledge, the first systematic analysis of a reasoning–reliability trade-off along the tool hallucination axis in tool-based agents. Regarding “other settings” to study the trade-off more systematically, this is an important direction, but much of the safety side of the space has already been explored by prior work [1,2,3], and factual hallucinations have also been widely studied [4]. We therefore focus on the under-explored and practically important case of tool hallucinations in agents, and view extending our framework to a broader set of reliability tasks as natural future work.
>
> ---
>
> **References**
>
> [1] Chegini, A., Kazemi, H., Souza, G., Safi, M., Song, Y., Bengio, S., Williamson, S., & Farajtabar, M. (2025). *Reasoning’s razor: Reasoning improves accuracy but can hurt recall at critical operating points in safety and hallucination detection*. arXiv preprint arXiv:2510.21049.
>
> [2] Huang, T., Hu, S., Ilhan, F., Tekin, S. F., Yahn, Z., Xu, Y., & Liu, L. (2025). *Safety tax: Safety alignment makes your large reasoning models less reasonable*. arXiv preprint arXiv:2503.00555.
>
> [3] Li, A., Mo, Y., Li, M., Wang, Y., & Wang, Y. (2025). Are Smarter LLMs Safer? Exploring Safety-Reasoning Trade-offs in Prompting and Fine-Tuning. *arXiv preprint arXiv:2502.09673*.
>
> [4] Akbar, S. A., Hossain, M. M., Wood, T., Chin, S.-C., Salinas, E. M., Alvarez, V., & Cornejo, E. (2024). HalluMeasure: Fine-grained hallucination measurement using chain-of-thought reasoning. EMNLP 2024.

---

### Official Review · Reviewer_ZS8Q · 2025-11-02

**Soundness:** 3
**Presentation:** 3
**Contribution:** 2
**Rating:** 4
**Confidence:** 4

**Summary:**

This paper investigates a timely and important question for agentic LLMs. The authors introduce SimpleToolHalluBench, a diagnostic benchmark that isolates two abstention-centric settings: No-Tool-Available (NTA) and Distractor-Tool (DT). They found that both tool-based reasoning, non-agentic reasoning RL, and distillation from thinking models would increase tool hallucination downstream. Mechanistically, they report that post-RL, tool-related representations collapse (lower CKA similarity) in early/mid layers while in-distribution representations remain stable, and that linear discriminability between correct and hallucinated trajectories concentrates in late-layer residual streams. Finally, the authors propose to mitigate the issue by prompt engineering and DPO. However, prompt engineering offers limited gains, while DPO reduces hallucinations but significantly degrades tool-using utility.

**Strengths:**

1. Clear, timely, and focused research question with practical impact for tool-using agents.

2. Strong empirical story across three angles:
   - Tool RL increases hallucination alongside reward.
   - Non-tool (math) RL increases hallucination, supporting causality tied to reasoning reinforcement.
   - Method-agnostic evidence via distillation and inference-time thinking mode.

3. Well mechanistic analysis. Layer-wise CKA highlights asymmetric stability, and linear probes localize discriminative differences to late residual streams, aligning with residual-as-accumulator intuitions.

4. Alternative solutions to the problem, although the solutions seem to be inferior currently.

**Weaknesses:**

1. The benchmark “guarantees” queries are impossible without the specific tool and counts direct answers as hallucinations. I think this is a strong assumption. Some queries might be partially answerable (e.g., templated guidance), so the label policy could conflate helpfulness with hallucination.

2. The proposed benchmark would be a bit small, just containing 296 samples. Additionally, it is created with the help of LLMs. I think there should be more descriptions about how you perform the quality control on the dataset, since it is the foundation of the entire work.

3. Enabling “thinking mode” typically increases output length and verbosity. Longer outputs may disproportionately increase the chance of mentioning tools, biasing the judge toward flagging hallucination. It is better to have length-controlled comparisons or comparisons under the same tool calls.

4. The mitigation suite is limited, just attempting prompting and DPO. Can it be solved with abstention-aware RL objectives (reward for correctly refusing when tools are absent, penalties for fabrications)? I think there should be many potential ways to mitigate the issues.

**Questions:**

1. Are the conclusions robust across seeds, temperatures, and decoding strategies (top-p, beam)?

2. Are the CKA collapses localized to particular attention heads or MLPs? Any evidence of module-specific drift (e.g., early attention heads linked to tool schema tokens)?

3. Would partial layer freezing during RL preserve tool pathways and reduce hallucination while keeping math gains?

---

> ### Author Response · Authors · 2025-11-20
> **Author Response to Reviewer ZS8Q(Part 1)**
>
> We sincerely appreciate the time and effort you have devoted to reviewing our paper, as well as your recognition of the strengths in our work. Below are our point-by-point responses to the weaknesses and questions raised.
>
> **For `Weakness 1`**: In all Agent architecture, any text matching the tool-call template is automatically executed by the system. It is not treated as a natural-language answer. Therefore, if the agent outputs a tool-call template for a tool that does not exist, this is necessarily a hallucination, not “helpfulness.” Even well-intentioned attempts (“calling” a tool just to be helpful) would trigger an invalid API call and cause real system errors.  In our experimental settings, attempts like “I don‘t have that tool, but if the user can provide XXX tool, I can solve the problem by XXX” will not be considered as the hallucination(see Appendix A.4 about our prompt used in LLM-as-Judge). Therefore, our experimental results ensure that genuinely helpful, non-hallucinatory responses providing illustrative examples are not penalized.
>
> **For `Weakness 2`**: We thank the reviewer for raising these crucial questions about the benchmark's assumptions and construction quality. We agree that the reliability of the dataset is foundational to our work and would like to provide more details on our rigorous quality control process.
>
> **Tool Selection and Query Construction:**
> We began by selecting **349 diverse tools** from AgentSafetyBench, deliberately covering broad scenarios including web search, operating system operations, transportation/traffic, financial transactions, healthcare data, and scientific computing to ensure systematic coverage. For each tool, we generated **two distinct query types** using ChatGPT-4o:
>
> - **Explicit Invocation Queries**: The user directly names the required tool (e.g., "Please use the `get_restaurant_address` tool to find the nearest KFC"). This is used in the **No-Tool-Available (NTA)** setting where the system prompt contains **no tools** and does not instruct the model to use tools, testing whether the model can abstain from hallucinating non-existent tools.
>
> - **Implicit Requirement Queries**: The user describes a task that *necessitates* a specific tool without naming it (e.g., "Find the current occupancy rate for hotel X"), while the system prompt only provides an irrelevant distractor tool. This is used in the **Distractor-Tool (DT)** setting to test whether the model correctly identifies that available tools are insufficient.
>
> **Manual Quality Control:**
> After LLM generation, **all candidate queries were manually reviewed by two independent annotators, resulting in the final set of 296×2 tool-query pairs** (with a third reviewer resolving disagreements). We explicitly filtered out queries where the answer could be partially derived from the model's internal knowledge (e.g., common facts, generic advice), ensuring our hallucination labels reflect **genuine fabrication** rather than helpful, knowledge-based responses. We respectfully emphasize that these 296 tools provide representative coverage across functional domains, and the benchmark's design purpose is specifically to **evaluate tool hallucination tendencies** rather than comprehensively assess tool-calling capability. For this targeted diagnostic goal, this lightweight benchmark is sufficiently effective.
>
> **For `Weakness 3`**: We appreciate this important concern. To disentangle the effect of reasoning enhancement from mere output verbosity, we conducted a **length-controlled experiment** using Qwen3-8B's thinking mode.
>
> We used **vLLM's beam search** with `length_penalty` parameters to systematically vary output length while keeping the thinking mode enabled. The results clearly demonstrate that hallucination rate is **not** driven by output length:
>
> | Model Configuration                              | $R_{DT} (\\%)$ |Avg. Output Tokens|
> | ---------------------------------------------------- | -------- | ------------------ |
> | Qwen3-8B (Thinking Disabled)                         | 36.2     | 127                |
> | Qwen3-8B (Thinking Enabled, beam+length_penalty=0.7) | 55.7     | 139                |
> | Qwen3-8B (Thinking Enabled, beam+length_penalty=1.0) | 56.8     | 269                |
> | Qwen3-8B (Thinking Enabled, beam+length_penalty=1.5) | 56.4     | 382                |
>
> **Crucially, even when controlling the thinking mode's output length to be nearly identical to the non-thinking baseline (139 vs. 127 tokens), the hallucination rate remains dramatically higher (55.7% vs. 36.2%).** Furthermore, hallucination rate stays stable across a 2.5x variation in output length among thinking-enabled models (139–382 tokens), indicating that verbosity alone cannot explain the effect. This evidence strongly suggests that reasoning enhancement affects hallucination through **altered internal confidence and decision patterns**, not superficial verbosity. We will include this analysis in the revised manuscript.

---

> ### Author Response · Authors · 2025-11-20
> **Author Response to Reviewer ZS8Q(Part 2)**
>
> **For `Weakness 4`**: We appreciate the suggestion to explore additional mitigation strategies. However, we emphasize that our DPO implementation already **theoretically operationalizes the abstention-aware RL objective** you propose, making it a principled and strong baseline. DPO is a RL algorithm that optimizes a policy $\pi_\theta$ by maximizing a preference model derived from implicit reward differences: $L_{\text{DPO}} = -\mathbb{E}[\log \sigma(\beta_{\text{DPO}} \Delta_\theta)]$, where $\Delta_\theta$ encodes the relative preference margin between chosen and rejected responses under a frozen reference policy $\pi_{\text{ref}}$. This is mathematically equivalent to RL with a learned reward function that penalizes undesirable behaviors while preserving capability.
>
> In practice, our preference dataset is explicitly constructed to implement the abstention-aware reward structure suggested by the reviewer.
>
> - **Positive reward for correct abstention**: When tools are absent, the **chosen** response $y^+$ is an honest refusal: *"I cannot answer because [required tool] is unavailable."* The **rejected** response $y^-$ is the hallucination (fabricated tool calls/outputs). This directly teaches the model to recognize tool unavailability.
>
> - **Penalty for over-cautiousness**: When tools *are* available, we reverse the preference: $y^+$ is correct tool invocation, while $y^-$ is needless refusal. This prevents the policy from collapsing into passive behavior.
>
> **For `Question 1`**: We appreciate this important robustness check. Our experiments confirm that hallucination increases are consistent across different hyperparameters. Below are three experiments on Qwen2.5-7B-Instruct demonstrating stability:
>
> | Seed    | $R_{NTA}(\\%)$ | $R_{DT}(\\%)$ |
> | ------- | ------------- | ------------ |
> | Seed=0  | 34.5          | 55.4         |
> | Seed=24 | 34.5          | 54.7         |
> | Seed=42 | 34.8          | 54.7         |
>
> | Temperature | $R_{NTA}(\\%)$ | $R_{DT}(\\%)$ |
> | ----------- | ------------- | ------------ |
> | T=0.0       | 34.5          | 54.7         |
> | T=0.5       | 34.8          | 54.7         |
> | T=0.7       | 36.1          | 54.4         |
> | T=1.0       | 35.8          | 56.1         |
>
> | Beam Size (best_of) | $R_{NTA}(\\%)$ | $R_{DT}(\\%)$ |
> | ------------------- | ------------- | ------------ |
> | best of=1 (Greedy)  | 34.8          | 54.7         |
> | bes of=4           | 34.8          | 54.4         |
> | best of=8           | 34.5          | 54.7         |
>
> Across all configurations, hallucination rates vary by less than **3% absolute**, confirming the effect is not a stochastic artifact of specific seeds or decoding strategies.

---

> ### Author Response · Authors · 2025-11-20
> **Author Response to Reviewer ZS8Q(Part 3)**
>
> **For `Question 2`**: We thank the reviewer for the suggestion to probe whether the CKA collapse is localized to particular modules. First, we clarify that our layer-wise CKA curves are computed on the **residual stream**: for each layer ℓ, we take the post-block residual activations and measure the CKA similarity between the pre-RL and post-RL models. Thus, the “collapse’’ we highlight is a property of the integrated residual representation, after aggregating the contributions of all attention heads and the MLP in that block. To probe modules more directly, we additionally compute CKA on the **attention output** and **MLP output** of each layer before they are added back to the residual stream (**See Figure 6 in the appendix D.1**). We find that their layer-wise trends closely follow the residual-stream curve: both attention and MLP CKA decrease smoothly with depth, rather than showing sharp drops at a few isolated blocks. A natural explanation is that both sublayers take the previous residual stream as input, so any global drift in residual representations induced by Reasoning RL propagates through both modules. We do observe that MLP CKA is consistently lower than attention CKA, suggesting that MLPs are somewhat more strongly updated by RL, but this effect is again distributed across layers instead of being localized to a small set of heads or blocks.
>
> Second, regarding attributing hallucination-related effects to specific attention heads or MLPs, existing mechanistic work on hallucinations primarily operates at the level of **residual-stream features or directions**, rather than single heads or neurons. For instance, ReDeEP[1] analyzes how “knowledge FFNs’’ and “copying heads’’ contribute to hallucinations via their impact on the residual stream and then detects/mitigates hallucination by modulating these residual-stream contributions. Jiang et al. [2] study factual hallucinations through the evolution of token information along the residual stream over layers and build detectors using these residual-level dynamics. Suresh et al. (2025)[3] train sparse autoencoders on residual-stream activations and show that manipulating a small set of learned concepts in the residual stream can systematically increase or decrease hallucinations. Together, these studies support a view of hallucination as arising from **distributed circuits embedded in the residual stream**, rather than from a few anomalous heads or single MLP units. In line with this literature, and given that our attention/MLP CKA changes are smooth and globally distributed (with only a mild MLP–attention gap), we believe it is difficult and potentially misleading to ascribe the observed CKA collapse to particular attention heads or MLPs. We therefore characterize our finding as a residual-stream–level representational drift induced by Reasoning RL, and we view fine-grained circuit tracing at the modules level as important future work beyond the scope of this paper.
>
>
> **For `Question 3`**: We appreciate the suggestion to mitigate the trade-off between the utility and hallucination. **Layer-wise CKA analysis reveals that layers 0-3 exhibit the most precipitous representational collapse** for tool-related queries (CKA drops from 0.99 to 0.76 within these layers), while math reasoning representations remain stable. This asymmetry suggests early layers are specialized for encoding tool affordances and are disproportionately disrupted by reasoning RL's optimization pressures.
>
> To test this, we piloted **freezing layers 0-3** during GRPO training on GSM8K, preserving their pretrained representations while allowing later layers to adapt for mathematical reasoning:
>
> | Configuration     | $GSM8K\\ Reward$ | $R_{NTA}(\\%)$ | $R_{DT}(\\%)$ |
> | ----------------- | --------------- | ------------- | ------------ |
> | Full Training     | 0.90            | 43.6          | 82.1         |
> | Freeze Layers 0-3 | 0.82 (-0.08)    | 42.9 (-0.7)   | 80.4(-1.7)   |
>
> Partial freezing reduces tool hallucination modestly (~2% absolute) while preserving 91% of math performance. However, the limited gain suggests hallucination also emerges from **residual compounding** (as shown in Section 5.2). This pilot supports our mechanistic view that the issue is not localized to some special layers alone but arises from **systematic drift across the entire depth**, warranting more targeted interventions.
>
> ---
>
> **References**
>
> [1] Sun, Z., Zang, X., Zheng, K., Song, Y., Xu, J., Zhang, X., Yu, W., & Li, H. (2025). *ReDeEP: Detecting hallucination in retrieval-augmented generation via mechanistic interpretability*. ICLR 2025.
>
> [2] Jiang, C., Yu, M., Zhou, B., & Zhang, P. (2024). *On large language models’ hallucination with regard to known facts*. NAACL 2024.
>
> [3] Suresh, P., Stanley, J., Joseph, S., Scimeca, L., & Bzdok, D. (2025). *From noise to narrative: Tracing the origins of hallucinations in transformers*. NeurIPS 2025.

---

> ### Author Response · Authors · 2025-11-26
> **Kind Request for Your Response**
>
> Dear Reviewer ZS8Q,
>
> I hope this message finds you well. As the discussion period is nearing its end, with less than one week remaining, I wanted to ensure that we have addressed all of your concerns satisfactorily. If there are any additional points or feedback you would like us to consider, please let us know. Your insights are invaluable to us, and we are eager to address any remaining issues to improve our work.
>
> Thank you for your time and effort in reviewing our paper.

---

### Author Response · Authors · 2025-11-29
**General Response**

Dear Reviewers and Area Chairs,

We sincerely thank all reviewers for their valuable feedback. We are encouraged that reviewers recognized our work as "clear, timely, and focused" (ZS8Q), "clearly written with experimental results supporting claims from different perspectives" (rLkj), and providing "novel mechanistic insight" (4Xi5).

During the rebuttal period, we have addressed all major concerns through extensive additional experiments and clarifications. Below we summarize the key improvements.

---

## Key Improvements Made

### Expanded Model Scale and Generalizability

In response to concerns about model coverage, we extended evaluation to **state-of-the-art reasoning models up to 1T parameters**, including DeepSeek-R1-671B, Qwen3-235B, and Kimi-K2-1T. Results consistently show that reasoning-enhanced versions exhibit higher tool hallucination rates across all scales, confirming our findings are not artifacts of smaller models.

### Controlled Experiments Ruling Out Confounds

We addressed concerns about potential confounding factors:

- **Length control**: Even when output length is matched (~130 tokens), thinking-enabled models show 55.7% vs 36.2% hallucination rate, demonstrating the effect stems from reasoning enhancement, not verbosity.
- **Robustness**: Results remain stable (<3% variation) across different random seeds, temperatures, and decoding strategies.
- **Cross-framework/domain validation**: We clarified that ReCall uses GRPO as its underlying algorithm, so our SynTool and GSM8K experiments already constitute cross-domain validation with the same RL framework.

### Deepened Mechanistic Analysis

We provided finer-grained analyses:

- **Module-level CKA** (new Figure 6): Both attention and MLP outputs show smooth depth-wise CKA decreases rather than localized collapses.
- **Cross-domain CKA** (new Figure 7): Tool-specific RL does not preserve a "safe" in-distribution regime—both domains undergo comparable representational drift.
- **Statistical significance**: Linear probe discrimination scores show 0.14±0.02 in late-layer residual streams across five random seeds.

### Theoretical Framework

We provided a shared-representation perspective explaining the reliability-capability trade-off: when reasoning and tool-reliability objectives share the same representation space, optimizing one mechanically distorts the other—a standard multi-task interference effect consistent with our empirical findings.

---

## Clarifications on Reviewer Misunderstandings

We respectfully note that some concerns appear to stem from misunderstandings of our paper:

- **Reviewer 4Xi5's concern about "cross-framework/domain validation"**: As clarified in our response, ReCall is not a separate RL algorithm but a training framework built on GRPO. This was stated in the original submission, and our experiments (SynTool+ReCall vs. GSM8K+GRPO) already provide cross-domain validation under the same algorithmic framework.

- **Reviewer Zskr's concern that "the conclusion seems unsurprising"**: We respectfully disagree. As detailed in our response with extensive citations, recent literature shows contradictory findings on reasoning-reliability relationships, and the tool hallucination dimension in agentic settings has never been systematically studied. Our work fills this important gap.

- **Reviewer Zskr's concern about benchmark novelty**: SimpleToolHalluBench serves a fundamentally different purpose from capability benchmarks like API-Bank. Existing benchmarks measure whether agents *can* use tools; ours measures whether agents *should abstain*—a critical reliability dimension that no prior benchmark addresses.

---

## Closing Remarks

We have invested substantial effort in conducting additional experiments and providing detailed clarifications during the rebuttal period. We are grateful to Reviewer rLkj for engaging with our response and acknowledging that our revisions adequately addressed their concerns.

We understand that due to the recent OpenReview security incident, reviewers are no longer able to participate in further discussion. While we regret that this unfortunate event has prevented us from having a more thorough exchange with Reviewers ZS8Q, 4Xi5, and Zskr, we believe our comprehensive responses have thoroughly addressed all raised concerns. We kindly request the Area Chair to take our detailed rebuttals and additional experiments into full consideration when making the final decision.

Thank you again for your time and constructive comments.

Best regards,
The Authors

---

### Note · Authors · 2026-01-04

I have read and agree with the venue's withdrawal policy on behalf of myself and my co-authors.